# Recovering Time-Varying Networks From Single-Cell Data

## Abstract

Gene regulation is a dynamic process that underlies all aspects of human development, disease response, and other key biological processes. The reconstruction of temporal gene regulatory networks has conventionally relied on regression analysis, graphical models, or other types of relevance networks. With the large increase in time series single-cell data, new approaches are needed to address the unique scale and nature of this data for reconstructing such networks. Here, we develop a deep neural network, Marlene, to infer dynamic graphs from time series single-cell gene expression data. Marlene constructs directed gene networks using a self-attention mechanism where the weights evolve over time using recurrent units. By employing meta learning, the model is able to recover accurate temporal networks even for rare cell types. In addition, Marlene can identify gene interactions relevant to specific biological responses, including COVID-19 immune response, fibrosis, and aging, paving the way for potential treatments.

## 1 Introduction

Biological systems are dynamic, changing over time in response to various stimuli and events. To construct accurate models of biological activity during development, disease progression, treatment response, and other biological processes, it is essential to track their evolution over time (Bar-Joseph et al., 2012). Studying the *regulation* of these dynamic processes is key for understanding the underlying mechanisms that drive the response and for identifying potential interventions that can serve as cures for diseases (Silverman et al., 2020).

Much of the research in this area is focused on the reconstruction of regulatory networks (Karlebach & Shamir, 2008; Badia-I-Mompel et al., 2023). These networks comprise a subset of proteins known as transcription factors (TFs), which regulate the activity of all other genes and proteins within the cell. However, these gene regulatory networks (GRNs) are not static. Instead, both the active nodes (proteins) and the edges (genes) change over time (Gasch et al., 2000; Luscombe et al., 2004). To reconstruct such networks, researchers often integrate static data—such as the type of nodes in the network—with dynamic data, such as time series measurements of node activity (gene expression profiles). Early work in this area employed microarrays and ChIP-chip data (Lee et al., 2002; Wang et al., 2006; Blais & Dynlacht, 2005; Gilchrist et al., 2009) followed by time series next-generation RNA-seq data (Wang et al., 2009), and most recently, single-cell RNA-seq (scRNA-seq) data (Ding et al., 2022; Nguyen et al., 2021; Matsumoto et al., 2017).

Several computational methods have been proposed over the last two decades to reconstruct such dynamic GRNs (Bar-Joseph et al., 2003; Yosef et al., 2013; Schulz et al., 2012; Yan et al., 2021). Some of these methods utilized time-varying graphical models including Hidden Markov models, Markov random fields, and Dynamic Bayesian Networks (Ahmed & Xing, 2009; Song et al., 2009; Dondelinger et al., 2013; Zhu & Wang, 2015). Other approaches attempted to use regression or to learn temporal precision matrices using extensions of the graphical lasso algorithm (Wang et al., 2020; Hallac et al., 2017).

While such models successfully reconstructed some processes (Ahmed & Xing, 2009; Kim et al., 2012), they are less suitable for more recent types of data, most notably scRNA-seq time series. First, the larger size of the data presents a challenge for traditional graphical models. Also, prior methods do not directly account for the fact that multiple cells are profiled for each time point. Finally, prior

methods do not leverage larger models, such as neural networks, which have demonstrated significant performance improvements across various learning tasks (Ching et al., 2018; Angermueller et al., 2016).

Very recently, a few methods have been proposed for using deep learning to recover static GRNs (Shrivastava et al., 2022; Shu et al., 2021). However, they cannot be directly used to capture dynamic GRNs (i.e., enforcing learning between time points). Two recent approaches, Dictys and CellOracle (Wang et al., 2023b; Kamimoto et al., 2023) can infer dynamic GRNs, however, these methods depend on data types like ATAC-seq, which provides direct information about TF binding sites but is less prevalent and harder to obtain.

Beyond the realm of biology, the inference of dynamic graphs using neural networks has garnered significant attention. This problem has found applications in diverse domains, including information retrieval, molecular graphs, and traffic forecasting (Zhu et al., 2021; Zhang et al., 2020). While there are similarities between these problems and the dynamic GRN problem, there are also significant differences that make it hard to extend these methods for time series scRNA-seq. The problem of inferring temporal graphs is usually defined by recovering a series of graph adjacency matrices $\mathbf{A}_t \in \mathbb{R}^{n \times n}$ where $n$ is the number of nodes and each node is a $k$-dimensional feature vector. However, when dealing with scRNA-seq data, the problem becomes: given a gene expression matrix $\mathbf{X}_t \in \mathbb{R}^{c \times g}$, where $c$ is the number of cells (samples) and $g$ is the number of genes (features), we are interested in recovering gene networks $\mathbf{A}_t \in \mathbb{R}^{g \times g}$, i.e., *graphs of features* rather than nodes (cells).

In this paper, we present a novel deep learning framework that effectively addresses the challenges discussed above for reconstructing dynamic GRNs. Our contribution is three-fold. First, we demonstrate that existing deep learning methods for temporal graph structure learning can be adapted for scRNA-seq data analysis. To achieve this, we perform a *gene featurization* step by leveraging set-like architectures such as DeepSets or Set Transformers (Zaheer et al., 2017; Lee et al., 2019). Second, we construct dynamic graphs by applying a self-attention mechanism (Bahdanau et al., 2014) to these gene feature vectors. To model dynamics, we draw inspiration from EvolveGCN where a gated recurrent unit (GRU) evolves the weights of a graph neural network (Pareja et al., 2020). However, unlike EvolveGCN, our approach uses a GRU to evolve the weights of key and value projection matrices in the self-attention module. This allows for the construction of dynamic graphs that capture regulatory interactions over time. Lastly, GRNs are highly dependent on cell functions, hence, separate GRNs need to be learned for each cell type. A single scRNA-seq dataset may combine cells of multiple types, some of which are rare cell populations. To this end, we employ a model-agnostic meta-learning (MAML) (Finn et al., 2017) training procedure by treating each cell type as a "task" to be learned. With this approach, the model quickly adapts to tasks with few samples, enabling the reconstruction of dynamic graphs even for rare cell types.

We apply our **me**ta lea**r**ning approach for inferring tempora**l** g**ene** regulatory networks (Marlene) to three publicly available scRNA-seq datasets. The first is a time series SARS-CoV-2 mRNA vaccination dataset of human peripheral blood mononuclear cells (PBMCs) (Zhang et al., 2023). The second dataset is a human lung aging atlas from the Human Cell Atlas Project (Regev et al., 2017; Sikkema et al., 2023). The third dataset is from a study of lung fibrosis using a mouse lung injury model (Strunz et al., 2020). All three datasets incorporate several time points, thus enabling a longitudinal analysis of the relevant biological responses through the inference of dynamic, cell type-specific GRNs. As we show, our method is able to reconstruct accurate networks for these datasets, significantly improving upon prior methods proposed for this task.

## 2 METHODS

### 2.1 PROBLEM SETUP

Consider a gene expression matrix $\mathbf{X} \in \mathbb{R}^{c \times g}$ where $c$ is the number of cells and $g$ is the number of genes. In the human genome, $g$ varies from 25,000 to 30,000, while the number of cells could be between a couple thousand to a few million. In the setting of dynamic graphs, we assume the existence of a time point for each row (cell), leading to a time series $\widetilde{\mathbf{X}} := \{\mathbf{X}_1, \ldots, \mathbf{X}_T\}$ with $\mathbf{X}_t \in \mathbb{R}^{c_t \times g}$. Here, the number of cells $c_t$ may vary with $t$. We are interested in recovering a series of directed graphs $\widetilde{\mathcal{G}} := \{\mathcal{G}_1, \ldots, \mathcal{G}_T\}$ where each $\mathcal{G}_t = \{\mathcal{N}, \mathcal{E}_t\}$. The set of nodes is the set of genes, i.e., $\mathcal{N} = [g]$, and we assume this set is static over time. The dynamic edge sets

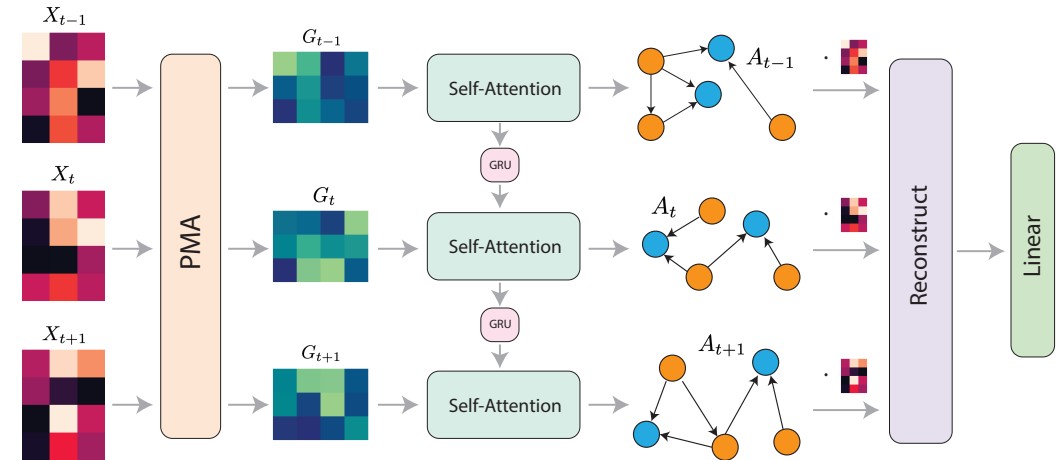

Figure 1: Overview of Marlene. Marlene takes as input gene expression data in the form of a cell-by-gene matrix. It then performs gene featurization via the pooling by multihead attention (PMA) mechanism which returns a gene feature matrix. This matrix is then inputted into a self-attention module to obtain a gene network in the form of an adjacency matrix. The weights of the self-attention module evolve from one time point to the next via a gated recurrent unit (GRU). The expression of transcription factors and the recovered graph are used to reconstruct the full gene expression vector. Finally, the reconstructed matrix is used to predict the cell type for the batch. The network is trained in a model-agnostic meta-learning fashion where each cell type is treated as a "task" to be learned, thus enabling the model to quickly adapt to cell types with low representation.

$\mathcal{E}_t = \{(u, v, w)\}_{u,v \in \mathcal{N}, w \in \mathbb{R}}$ denote directed weighted links between genes, where the source $u$ is the gene that regulates the expression of its target $v$, and $w$ is the strength of this relationship. The source nodes are called transcription factor genes (TFs).

We can alternatively characterize each graph $\mathcal{G}_t$ by the corresponding adjacency matrix $\mathbf{A}_t \in \mathbb{R}^{g \times g}$. Denote $\widetilde{\mathbf{A}} := \{\mathbf{A}_1, \ldots, \mathbf{A}_T\}$. Since TFs control the expression of their target genes, the underlying GRNs should, in principle, allow the recovery of the full expression profile for a cell. In other words, $\widetilde{\mathbf{X}} = f(\widetilde{\mathbf{X}}^{\text{TF}}, \widetilde{\mathbf{A}})$ where $\widetilde{\mathbf{X}}^{\text{TF}}$ denotes the expression of all TFs. The function $f$ is unknown as it involves intricate interactions among genes, including combinatorial effects. For instance, certain scenarios exist where TFs cooperate with each-other to activate a gene, while in other instances, the activation requires some TFs to be active and others to be inactive (Wise & Bar-Joseph, 2015).

In existing deep learning literature, $f$ is sometimes modeled using autoencoders (Seninge et al., 2021; Shu et al., 2021; Wang et al., 2023a). However, reconstructing the full gene expression vector is challenging as the data is extremely sparse and conventional reconstruction losses, such as mean squared error, tend to emphasize overall averages. Since GRNs are dependent on cell function, we hypothesize that simplifying the problem by predicting cell types may improve the accurate recovery of GRNs. In other words, given a temporal batch of cells of the same type $\widetilde{x}^{\text{TF}} \in \mathbb{R}^{T \times \text{batch size} \times |\text{TFs}|}$, we consider the classification problem $y = f(\widetilde{x}^{\text{TF}}, \widetilde{\mathbf{A}})$ where $y$ is the known cell type label for the batch. Finally, the task of learning $\widetilde{\mathbf{A}}$ given a batch $\widetilde{x}$ becomes

$$\arg\min_{\widetilde{\mathbf{A}}} \text{CrossEntropyLoss}(y, f(\widetilde{x}^{\text{TF}}, h(\widetilde{x}))), \qquad \widetilde{\mathbf{A}} := h(\widetilde{x}) \qquad (1)$$

for choices of functions $f$ and $h$ where $h$ uses the expression data to obtain the adjacency matrices.

## 2.2 Architecture of Marlene

In this work, we propose a neural network architecture called Marlene that effectively learns dynamic GRNs (Figure 1). Marlene consists of three main steps. The first two steps address the choice of $h$ in (1), while the last addresses the choice of $f$.

In the first step, we apply a gene featurization step by treating a batch of cells as a set of elements. The DeepSet architecture introduces a pooling operator that allows the neural network to be invariant to the order of input samples, effectively treating the input as a set (Zaheer et al., 2017). Similarly, the Set Transformer architecture is designed to process sets of data via attention-based operators that are permutation invariant (Lee et al., 2019). Specifically, the pooling by multihead attention (PMA) aggregation scheme introduced in Set Transformers outputs a matrix of $k$ vectors $\mathbf{H} \in \mathbb{R}^{k \times g}$ for an arbitrary input set $\mathbf{X} \in \mathbb{R}^{c \times g}$. Each of the $k$ output vectors has a specific meaning such as statistics of the input data. By applying the PMA operator to a temporal batch of cells, we obtain a gene-by-feature matrix $\mathbf{G} = \mathbf{H}^{\top} \in \mathbb{R}^{g \times k}$ that encodes information about the cells from each time point. PMA consists of a multihead attention block (MAB) where the input $\mathbf{X}$ consists of key vectors, and the query is a learnable set of $k$ vectors $\mathbf{S} \in \mathbb{R}^{g \times k}$. In this work, we use a shared PMA layer for all time points assuming that the specific key statistical properties are invariant (though their value obviously changes for different time points). Given $\widetilde{x} = [x_1, \ldots, x_T]$, we have

$$\widetilde{\mathbf{G}} = \text{PMA}(\widetilde{x})^{\top} := \text{MAB}(\mathbf{S}, \widetilde{x})^{\top} := (\widetilde{\mathbf{M}} + \text{rFF}(\widetilde{\mathbf{M}}))^{\top} \tag{2}$$

where $\mathbf{M} = \mathbf{S} + \text{Multihead}(\mathbf{S}, x, x) \in \mathbb{R}^{g \times k}$ and rFF is a row-wise feedforward layer. For completeness, these operations are defined in the appendix.

Note that if cells of different types are mixed in the same batch, the statistics derived by the PMA step may not capture cell type-specific information. Consequently, in a single batch, we only include cells of one type.

In the second step, Marlene learns temporal adjacency matrices using a self-attention mechanism. To model dynamics, we draw inspiration from EvolveGCN which performs model adaptation using a GRU (Pareja et al., 2020). Unlike EvolveGCN, which uses the GRU to update the weights of a graph convolution layer, we use a GRU to evolve the key and query projection weights of the self-attention module. Since most time series we deal with contain very few time points, a GRU should suffice and not suffer from vanishing gradient problems. Similar to EvolveGCN, we apply a summarization step via top $k$ pooling to reduce the gene feature matrix to a square matrix for the GRU (Appendix).

More precisely, we initialize self-attention weights $\mathbf{W}_0^Q, \mathbf{W}_0^K \in \mathbb{R}^{k \times k}$ and two recurrent units $\text{GRU}_Q, \text{GRU}_K$. Given the time sequence of gene feature matrices $\widetilde{\mathbf{G}}$ obtained from the previous step, temporal adjacency matrices are constructed in the following recurrent fashion for all $t \in [T]$:

$$\mathbf{Z}_t = \text{TopK}(\mathbf{G}_t) \in \mathbb{R}^{k \times k} \tag{3}$$

$$\mathbf{W}_t^Q = \text{GRU}_Q(\mathbf{Z}_t, \mathbf{W}_{t-1}^Q), \quad \mathbf{W}_t^K = \text{GRU}_K(\mathbf{Z}_t, \mathbf{W}_{t-1}^K) \tag{4}$$

$$\mathbf{Q}_t = \mathbf{G}_t \mathbf{W}_t^Q, \quad \mathbf{K}_t = \mathbf{G}_t \mathbf{W}_t^K \tag{5}$$

$$\mathbf{A}_t = \text{softmax}\left(\frac{\mathbf{Q}_t \mathbf{K}_t^{\top}}{\sqrt{k}}\right). \tag{6}$$

Here, $\mathbf{W}_t^Q$ and $\mathbf{W}_t^K$ serve as hidden states for the respective GRUs. The GRUs dynamically adapt self-attention weights, influencing which TFs specific genes should attend to in subsequent time steps. Consequently, the evolution of these weights is constrained. We also restrict the columns of $\mathbf{A}_t$ (i.e., sources) to $p$ known TFs in the TRRUST database (Han et al., 2018) which greatly reduces the number of parameters to be learned. Therefore, in our implementation $\mathbf{A}_t \in \mathbb{R}^{g \times p}$.

Next, we perform a gene expression reconstruction step based on the expression of TFs and the inferred adjacency matrices. This is followed by any number of fully connected layers with nonlinear activation functions $\sigma$. Finally, we sum across output vectors to obtain a logit vector with the same dimension as the number of cell types in the data:

$$\tilde{y} = \text{Pool}(\text{Linear}(\ldots \sigma(\text{Linear}(\widetilde{x}^{\text{TF}} \widetilde{\mathbf{A}}^{\top})))). \tag{7}$$

Network depth can be introduced at all three levels by stacking MAB layers during gene featurization, stacking GRUs, or stacking linear layers at the end.

### 2.3 META LEARNING FOR RARE CELL TYPES

ScRNA-seq datasets often originate from biological samples that exhibit cellular heterogeneity, potentially containing multiple distinct cell types. Some of these cell subpopulations are rare and are

Table 1: Time series scRNA-seq datasets used in this study.

| Dataset | Number of | | | | Metadata | |
| | Cells | Genes[1] | TFs | Cell Types | Time Points | Sample |
| --- | --- | --- | --- | --- | --- | --- |
| SARS-CoV-2 | 113,271 | 1899 | 556 | 7 | d0, d2, d10, d28 | PBMCs (human) |
| HLCA | 27,953[2] | 2433 | 674 | 11 | Ages $< 35, 35 - 50, \geq 50$ | lung (human) |
| Fibrosis | 22,758 | 1217 | 433 | 6 | PBS, d3, d7, d10, d14, d21, d28 | lung (mouse) |

[1] Only showing the number of genes overlapping with the TRRUST database.

[2] We randomly sampled cells from 11 cell types.

represented by a small number of cells in the sample (Jindal et al., 2018). Since we are concerned with the discovery of cell type-specific temporal GRNs, learning such large graphs for these rare cell types may not be feasible and lead to overfitting. Since many interactions are shared across cell types (Chasman & Roy, 2017), we employ the model-agnostic meta-learning framework (MAML) (Finn et al., 2017). MAML is specifically designed to enable neural networks to adapt to novel tasks with limited training samples (i.e., few shot learning). By treating each cell type as a "task", the MAML training paradigm facilitates the recovery of dynamic graphs for rare cell types. We begin by adapting model parameters through multiple optimization steps using a batch of support examples (cells). These adapted parameters are then evaluated on a separate set of query cells, followed by a meta-update.

During the adaptation step, we perform gradient descent, while for the meta-update, we employ the Adam optimizer (Kingma & Ba, 2014). During training, gradient clipping proves crucial to prevent overfitting of the MAML adaptation step to the cell type under consideration.

## 3 EXPERIMENTS

To validate our approach, we use three public scRNA-seq datasets (Table 1): a human SARS-CoV-2 mRNA vaccination dataset, a lung aging atlas (The Human Lung Cell Atlas—HLCA), and a mouse lung fibrosis dataset (Zhang et al., 2023; Sikkema et al., 2023; Strunz et al., 2020). To assess the quality of the inferred networks, we draw upon two databases of TF-gene regulatory interactions, which have been curated from the scientific literature—TRRUST and RegNetwork (Han et al., 2018; Liu et al., 2015). For the human genome, TRRUST contains 8427 unique validated regulatory edges, while RegNetwork contains 150,405. Note that certain edges lack a corresponding TF or gene in the expression data, so the numbers used for the analysis are smaller. We used only the genes that were present in TRRUST for all three datasets. For the mouse lung dataset we used the corresponding mouse networks for both databases. To match the number of links in these databases, we selected the top 2% of edges for all methods. For Marlene, this was done by sparsifying the self-attention matrix to retain only the top scoring edges. The significance of the overlap was carried out via Fisher's exact test (Fisher, 1922). All $p$-values were corrected for multiple testing using the Benjamini-Hochberg procedure (Benjamini & Hochberg, 1995).

We compare Marlene against several popular static gene regulatory network inference methods included in the BEELINE benchmark (Pratapa et al., 2020) and beyond, such as PIDC, GENIE3, GRNBoost2, SCODE, and DeepSEM (Chan et al., 2017; Huynh-Thu et al., 2010; Moerman et al., 2019; Matsumoto et al., 2017; Shu et al., 2021), which are applied independently to each time point. DeepSEM is a deep generative model based on structural equation modeling. We also compare against time-varying graphical lasso (TVGL)[1], a method that models temporal precision matrices (Hallac et al., 2017), and to a deep neural network that utilizes the S4 module (GraphS4mer) (Tang et al., 2022; Gu et al., 2021).

During inference, we obtain multiple $A_t$ for different batches and average them. We train Marlene with a batch size of 16 cells and also use 16 seeds in the PMA layer. For MAML, we use 5 inner steps. The model with the lowest loss is selected for GRN inference. For the meta-update, we use a decaying learning rate starting with $1\mathrm{e}{-4}$, while for the inner step we use $1\mathrm{e}{-3}$ for both datasets. Experiments were performed using an NVIDIA RTX 3060 and took only a few minutes per run.

---

[1] We used the implementation of `https://github.com/fdtomasi/regain`

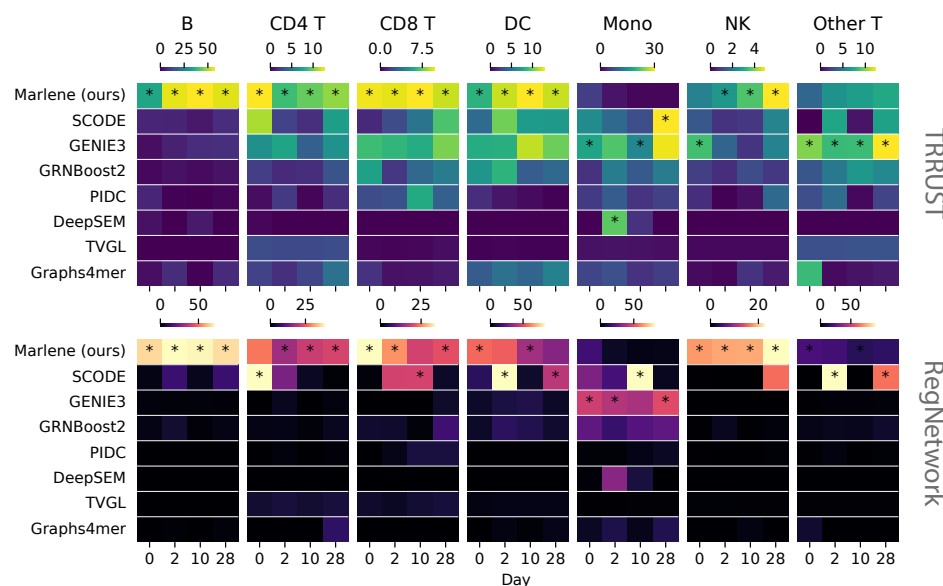

Figure 2: Overlap analysis of the SARS-CoV-2 vaccination dataset. Showing $-\log_{10}(\text{FDR})$ values from a Fisher's exact test measuring the overlap between predicted TF-gene interactions in reconstructed networks and two TF-gene interaction databases, TRRUST (top) and RegNetwork (bottom). Cell types are shown as columns. Best performing method is starred.

## 3.1 CASE STUDY 1: SARS-COV-2 VACCINATION

The SARS-CoV-2 vaccination dataset consists of peripheral blood mononuclear cells (PBMCs) from six healthy donors at four time points (days 0, 2, 10, and 28) (Zhang et al., 2023). Day 0 samples were obtained before vaccination. We removed the "Other" cell type group and kept the remaining seven. These include B cells, dendritic cells (DC), monocytes (Mono), natural killer cells (NK), and various types of T cells.

### 3.1.1 MARLENE RECOVERS ACCURATE GENE REGULATORY NETWORKS

Analysis results using Marlene and prior methods is presented in Figure 2. As can be seen, Marlene outperformed competing methods in the dynamic GRN inference task for 5 of the cell types, yielding statistically significant results across time points. Specifically, for B cells, Marlene successfully identified more than 800 regulatory links within the RegNetwork framework at each time point (FDR $\leq$ 1e$-$67), surpassing the performance of the second-best method, SCODE, which detected 579 links (day 2, FDR $\leq$ 1e$-$15). Analogous findings were observed for natural killer cells, where Marlene identified over 600 RegNetwork links at each time point (FDR $\leq$ 1e$-$18). In comparison, the second-ranking method, SCODE, showed a significant overlap for only one time point (day 28). Upon examining the TRRUST database, we observed less pronounced differences in the results. Nonetheless, Marlene obtained higher overlap for 5 out of 7 cell types followed by GENIE3, which performed well for monocytes and the "Other T" category.

### 3.1.2 MARLENE RECOVERS REALISTIC DYNAMIC TRANSITIONS

The analysis so far has primarily focused on individual time points. Next, we turned our attention to assessing the quality of graph transitions between consecutive time points. Specifically, we examined whether the learned graphs demonstrated smooth transitions over time. To evaluate this, we computed the intersection-over-union (IoU) score for edges between time points $t$ and $t + 1$ (Figure 3a). Notably, our findings revealed that for most cell types, Marlene exhibited the lowest IoU score during the initial period (days $0 \rightarrow 2$), followed by higher scores during days $2 \rightarrow 10$, and $10 \rightarrow 28$. This pattern aligns with our expectations, as variations in gene expression are likely

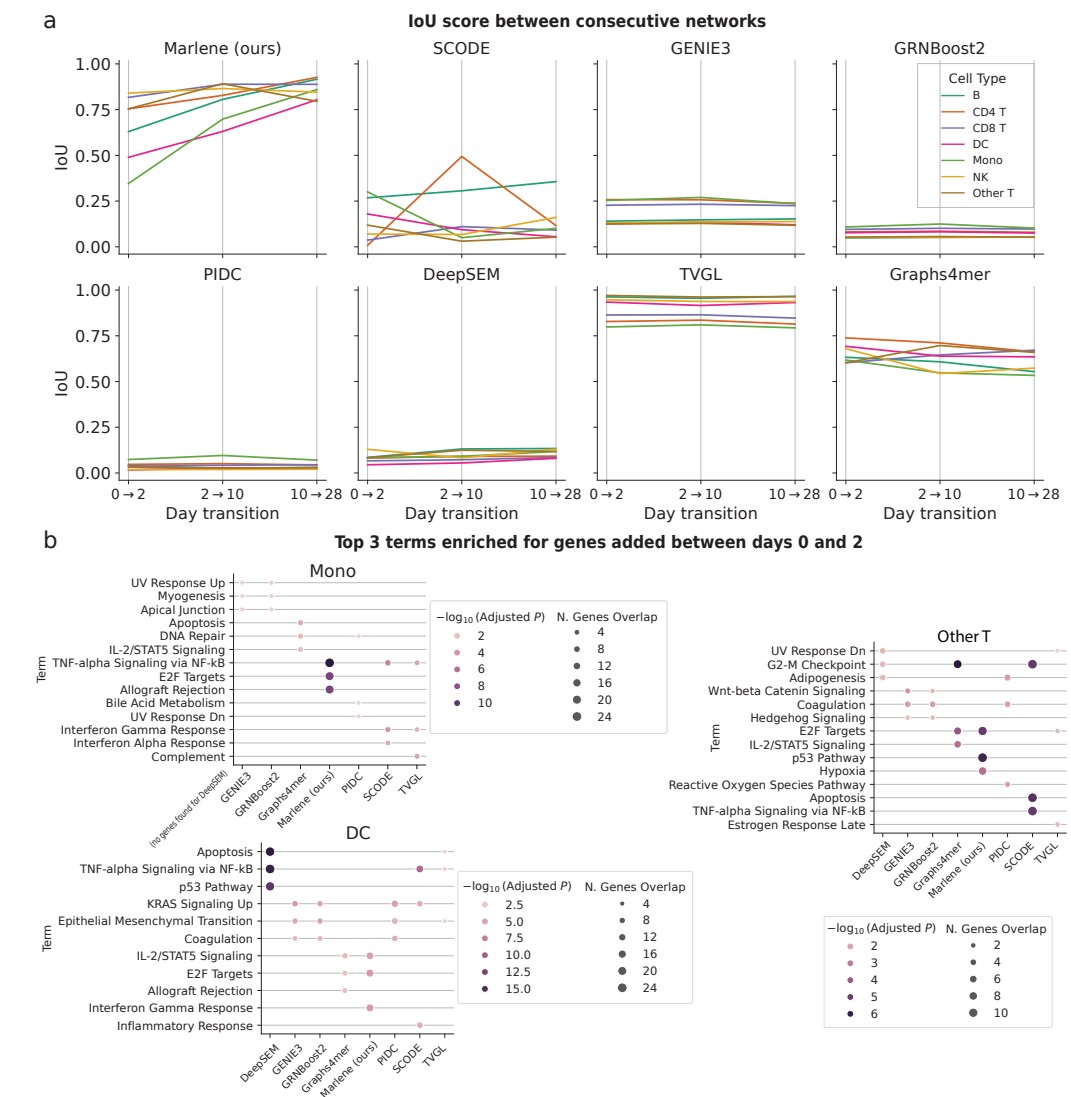

Figure 3: Temporal analysis of the predicted gene regulatory networks for the SARS-CoV-2 vaccine dataset. (a) Intersection-over-union (IoU) scores between consecutive graphs. (b) For each method, top 3 MSigDB terms enriched for genes that were regulated at day 2 but not day 0.

to be most pronounced during the early post-vaccination period (days $0 \rightarrow 2$). From the methods we compare against, GENIE3, GRNBoost2, SCODE, PIDC, and DeepSEM exhibited significantly lower IoU scores across all temporal transitions, likely due to their lack of dynamic modeling and the fact that they were ran independently per time point. TVGL, on the other hand, showed high IoU scores which remained close to constant over time. Finally, Graphs4mer displayed a reduction in IoU scores over time, which is unlikely given the expected immediate immune response.

Next, we sought to assess the quality of the TF-gene regulatory links added between time points. For brevity, we focused specifically on the initial temporal transition (days $0 \rightarrow 2$), as this period is likely to witness a more significant biological response. For each cell type, we took note of all the genes that were regulated by some TF at day 2 but not at day 0. Using this set of genes $z$, we performed gene set enrichment analysis (GSEA) (Subramanian et al., 2005; Fang et al., 2023) using the molecular signatures database (MSigDB) (Liberzon et al., 2015). Through permutation tests, GSEA assigns an enrichment score (ES) to $z$ reflecting its overrepresentation within the MSigDB gene set collection. We found that for many cell types, genes added by Marlene at day 2 greatly

overlapped with COVID-19 and SARS-CoV-2 related gene sets. For instance "Interferon Gamma Response", which was identified as a SARS-CoV-2 antiviral response (Hilligan et al., 2023), was significantly enriched in dendritic cells (15 genes, FDR = 1e−6). Similarly, "TNF-alpha Signaling via NF-kB"—a pathway involved in the immune response and inflammation (Hayden & Ghosh, 2011)—was enriched in several cell types, as well as processes such as "Apoptosis" (cell death) and "p53 Pathway" (inhibits replication of infected cells) (Elmore, 2007; Harris & Levine, 2005). Other methods, while being enriched for relevant terms, showed a smaller gene overlap for these types (Figure 3b) or were not consistent across cell types (e.g., DeepSEM, SCODE).

Overall, these results suggest that Marlene is able to capture both known TF-gene links, but also genes that are relevant to the response being studied.

## 3.2 CASE STUDY 2: AGING AND SENESCENCE IN THE LUNG

The Human Lung Cell Atlas (HLCA) is a large data integration effort by the Human Cell Atlas Project (Sikkema et al., 2023; Regev et al., 2017). This data combines scRNA-seq samples from 107 individuals spanning an age range of 10 to 76 years, making it particularly attractive for studying aging and senescence (a form of aging characterized by the absence of cell division) (van Deursen, 2014; SenNet Consortium, 2022).

We split the atlas into three age groups at 35 and 50 years old, thus forming a pseudotime series of length 3. We removed smokers from the dataset as these will likely confound the results. To accommodate the data in the GPU, we randomly selected cells from 11 cell types, including type II pneumocytes, endothelial cells, and monocytes.

Similar to the vaccination dataset, we begin the analysis by evaluating the set of regulatory links using the TRRUST and RegNetwork databases. For this dataset we find that Marlene and SCODE are the top two performing methods (Figure 4a). For some of the cell types, Marlene achieves significant results, recovering more than 1000 RegNetwork links (classical monocytes, FDR = 1e−76). Even for cell types with fewer cells, such as non-classical monocytes (with only 138 cells for the second age group), Marlene still recovered more than 800 known TF-gene links for each transition (FDR ≤ 1e−27). SCODE performed well for some cell types such as CD1c-positive myeloid dendritic cells and CD4-positive, alpha-beta T cells. For all other methods, the overlap was smaller (Appendix Figure 6a). Note that while SCODE is comparable for the static network (single time point) inference task, it does not utilize dynamic information.

We next examined the ability of different methods to capture the dynamics of the biological processes. For this, we looked at graph transitions. IoU scores show that only the temporal methods (Marlene, TVGL, Graphs4mer) capture the smooth temporal transition between time points, while other methods, including SCODE, achieve low IoU scores (Appendix Figure 6b). We performed GSEA using Jensen Diseases gene set to see if genes added by Marlene in these transitions were enriched for any age-related diseases (Pletscher-Frankild et al., 2015; Grissa et al., 2022). We found that Marlene added genes are enriched for several diseases such as arthritis, lung disease, and coronary artery disease. Other dynamic baselines were also enriched for relevant terms, but contained fewer marker genes (Figure 4b).

Finally, we also investigated whether the genes regulated at different age groups were enriched for senescence. Cellular senescence refers to a permanent arrest of cell division triggered by the accumulation of DNA damage (Suryadevara et al., 2024). The absence of cell division can detrimentally impact tissue regeneration and repair, thereby contributing to various age-related diseases. Here, we use the SenMayo gene set which contains 125 genes reported to be enriched for senescence (Saul et al., 2022). Only 81 of these genes overlapped with our data. We found that for 4 cell types, there was an increase in SenMayo gene regulation at the oldest age group (age > 50), suggesting that senescent cells accumulate with age as hypothesized (Figure 5).

## 3.3 CASE STUDY 3: FIBROSIS IN A MOUSE LUNG INJURY MODEL

Next, we evaluated whether Marlene could perform effectively across different species by analyzing a dataset from a mouse model of lung injury induced by the chemotherapeutic agent bleomycin (Strunz et al., 2020). The dataset included seven time points: one pre-treatment and six post-

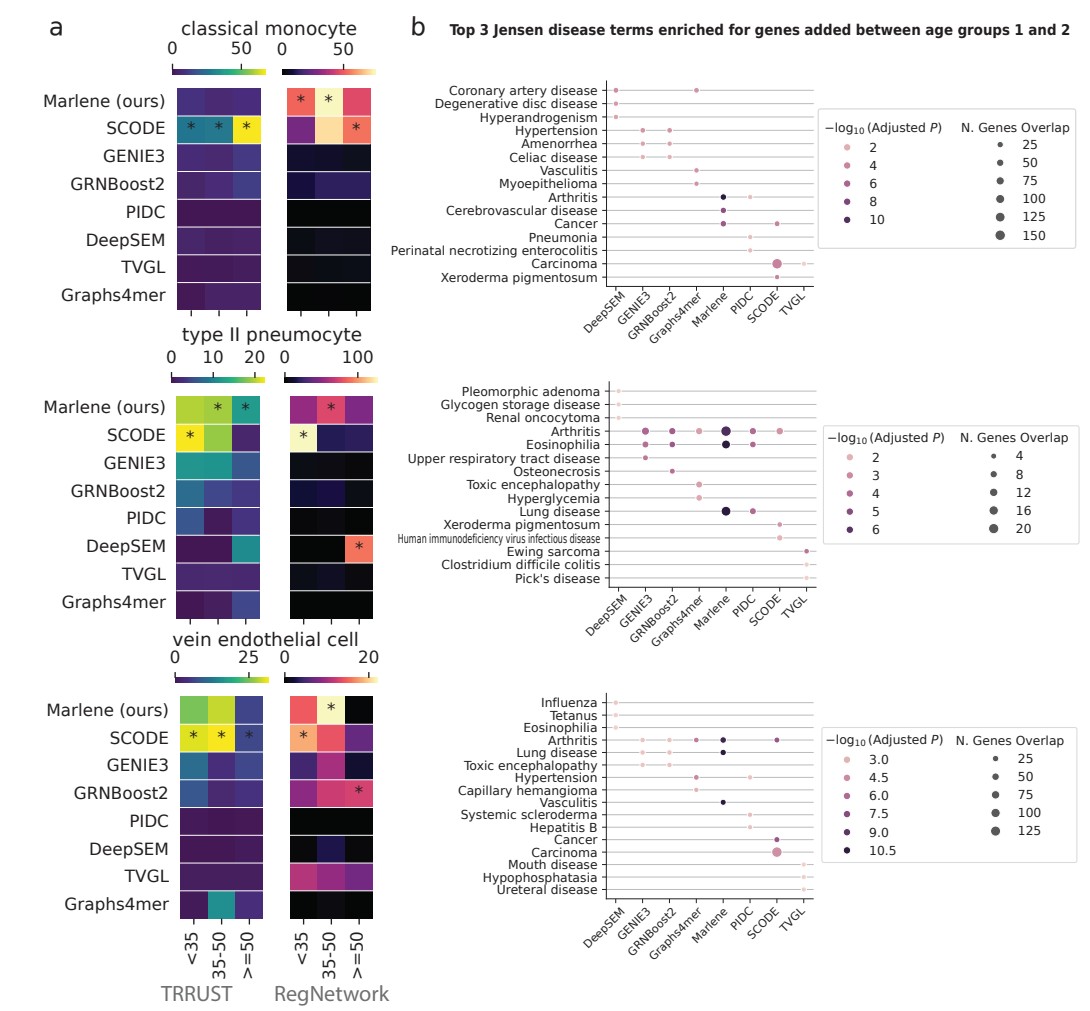

Figure 4: Results on the HLCA dataset. (a) FDR corrected $p$-values of Fisher exact tests reflecting the number of links that overlap with TRRUST and RegNetwork databases. (b) Top 3 Jensen Diseases terms enriched for genes added between the first and second age group.

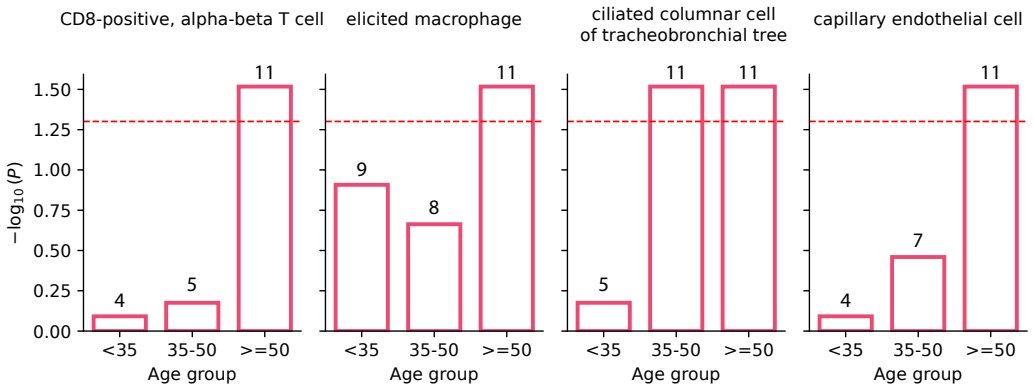

Figure 5: Enrichment for senescence using the SenMayo set. For 4 cell types, there was statistically significant enrichment for the oldest age group. We only used the top 200 regulated genes.

treatment intervals. After filtering out cell types with low representation and genes with low counts, we retained six cell types, including B cells, T cells, and macrophages.

In this analysis, Marlene outperformed competing methods in four of the six cell types when benchmarked against the RegNetwork database, specifically in alveolar epithelial cells, dendritic cells, endothelial cells, and macrophages. For T cells, TVGL showed slightly better performance. When evaluated against the TRRUST database, SCODE performed well in four cell types, while Marlene surpassed it in the remaining two. The differing results between two databases may reflect their incomplete coverage, highlighting the need for further refinement.

Finally, all static baselines, including SCODE, showed low IoU scores across time points, indicating their inability to capture temporal evolution. In contrast, Marlene, showed increasing IoU scores over time, suggesting ongoing lung regeneration following treatment which slowly stabilizes. Figures illustrating these findings are provided in the appendix.

## 4 DISCUSSION

Gene regulation is a dynamic process that underlies all biological systems. Understanding which TFs regulate which genes, and when this regulation occurs, provides insights into these dynamic processes which can lead to better treatment options. For instance, understanding what TF-gene links are disrupted could help researchers discover drugs targets for specific TF-gene connections.

To improve on current methods for reconstructing time varying regulatory networks, we use the expressive capabilities of deep neural networks to model the dynamic regulation of genes. Specifically, we focused on inferring dynamic networks from scRNA-seq data.

Our proposed method, Marlene, constructs dynamic graphs from time series data. Marlene begins with a set pooling operator based on PMA to extract gene features. These gene features are then used to construct dynamic graphs via a self-attention mechanism. The weights of the self-attention block are updated through the use of GRUs. Additionally, by employing MAML, we help Marlene uncover graphs even for rare cell types However, Marlene optimizes the prediction of cell type label rather than gene expression. As such, Marlene is not currently equipped to determine the impact of perturbations including gene knockouts or overexpression experiments. Exploring the integration of causal inference capabilities into Marlene represents a promising direction for future research.

We demonstrated the effectiveness of Marlene in recovering dynamic GRNs using three datasets: a SARS-CoV-2 vaccination dataset, a lung aging atlas, and a mouse dataset of fibrosis. In all three datasets, Marlene successfully identified many validated TF-gene links from the TRRUST and RegNetwork databases across various cell types. It also accurately modeled the temporal dynamics of these connections. Some prior methods ignored the temporal aspect, leading to little similarity between consecutive networks. Other methods integrated all time points together, leading to very similar networks for each time point. In contrast, Marlene accurately recovered the variation dynamics, which is often characterized by strong rewiring following treatment that later stabilizes. In addition, Marlene identified many relevant edges. For instance, in the lung aging data, several dynamic edges were enriched for age-related diseases, such as arthritis. Meanwhile, in the SARS-CoV-2 data, these dynamic links were enriched for immune response processes. Prior methods captured some known edges, however, the overall results were less significant. By providing better models to explain disease and vaccine response, researchers can zoom in on the specific mechanisms targeted which in turn can lead to better treatments. Code will be made publicly available on publication.

## 5 LIMITATIONS

While successful, Marlene has a few limitations. The datasets we used in this study, while typical for scRNA-seq time series, consisted of only a few time points. For longer sequences, the GRU operation may suffer from vanishing gradient problems (Pascanu et al., 2012). In such scenarios, the S4 module may be preferred as it has been shown to model long sequences better than traditional GRUs (Gu et al., 2021). In addition, using a large number of genes for training, results in quadratic growth in memory consumption due to the need to store adjacency matrices. This led us to restrict the set of genes for each of the two studies. A more efficient implementation or alternative approaches such as FlashAttention (Dao et al., 2022) can lead to better ability to utilize all genes profiled.

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

# A  APPENDIX

## A.1  SET TRANSFORMER OPERATIONS

We redefine the Multihead and rFF operations from Set Transformers (Lee et al., 2019) to the ones used for Marlene here.

First, we define the **Attention** operation. Let $Q \in \mathbb{R}^{k \times g}$ be the query matrix of $k$ elements and $g$ dimensions. The Attention operation used for MAB is

$$\text{Attention}(Q, K, V) = \text{softmax}\left(\frac{QK^\top}{\sqrt{g}}\right)V \tag{8}$$

where the key and value matrices are $K, V \in \mathbb{R}^{c \times g}$. Next, the Multihead attention operation with $h$ heads (Vaswani et al., 2017) is given by

$$\text{Multihead}(Q, K, V) = \text{concat}(O_1, \ldots, O_h)W^O \tag{9}$$

where $O_j = \text{Attention}(QW_j^Q, KW_j^K, VW_j^V)$ for weight matrices $W_j^Q, W_j^K, W_j^V \in \mathbb{R}^{g \times g/h}$ and $W^O \in \mathbb{R}^{g \times g}$ (these matrices are not to be confused with self-attention weights used consequently for Marlene). In our implementation, $k$ is the number of seeds or output vectors used for the PMA layer. This is a hyperparameter that corresponds to the number of "statistic" vectors we expect to learn from data. Finally, rFF is a feedfoward layer such as a linear layer.

## A.2  EVOLVEGCN OPERATIONS

Here, we introduce the GRU and topK pooling operations used in the second step of Marlene.

The topK pooling operation is needed to summarize nodes into $k$ representative ones (Cangea et al., 2018; Pareja et al., 2020). Here $k$ is the same as the number of seeds used for PMA. Given an input $\mathbf{G} \in \mathbb{R}^{g \times k}$ and a learnable vector $q$, the TopK operation performs the following steps:

$$\rho = \frac{\mathbf{G}q}{\|q\|}$$
$$i = \text{Top-k-indices}(\rho)$$
$$\mathbf{Z} = [\mathbf{G} \odot \tanh(\rho)]_i.$$

At time step $t$, given a pooled matrix $\mathbf{Z}_t$ and hidden state $\mathbf{W}_{t-1}$ (i.e., self-attention weights $\mathbf{W}_{t-1}^Q$ or $\mathbf{W}_{t-1}^K$), the standard GRU operation is:

$$r_t = \sigma(M_{ir}\mathbf{Z}_t + b_{ir} + M_{hr}\mathbf{W}_{t-1} + b_{hr})$$
$$z_t = \sigma(M_{iz}\mathbf{Z}_t + b_{iz} + M_{hz}\mathbf{W}_{t-1} + b_{hz})$$
$$n_t = \tanh(M_{in}\mathbf{Z}_t + b_{in} + r_t \odot (M_{hn}\mathbf{W}_{t-1} + b_{hn}))$$
$$\mathbf{W}_t = (1 - z_t) \odot n_t + z_t \odot \mathbf{W}_{t-1}$$

where $\sigma$ is the sigmoid function and $\odot$ is the Hadamard product. See also Paszke et al. (2019).

### A.3 SUPPLEMENTARY FIGURE FOR THE HLCA DATASET

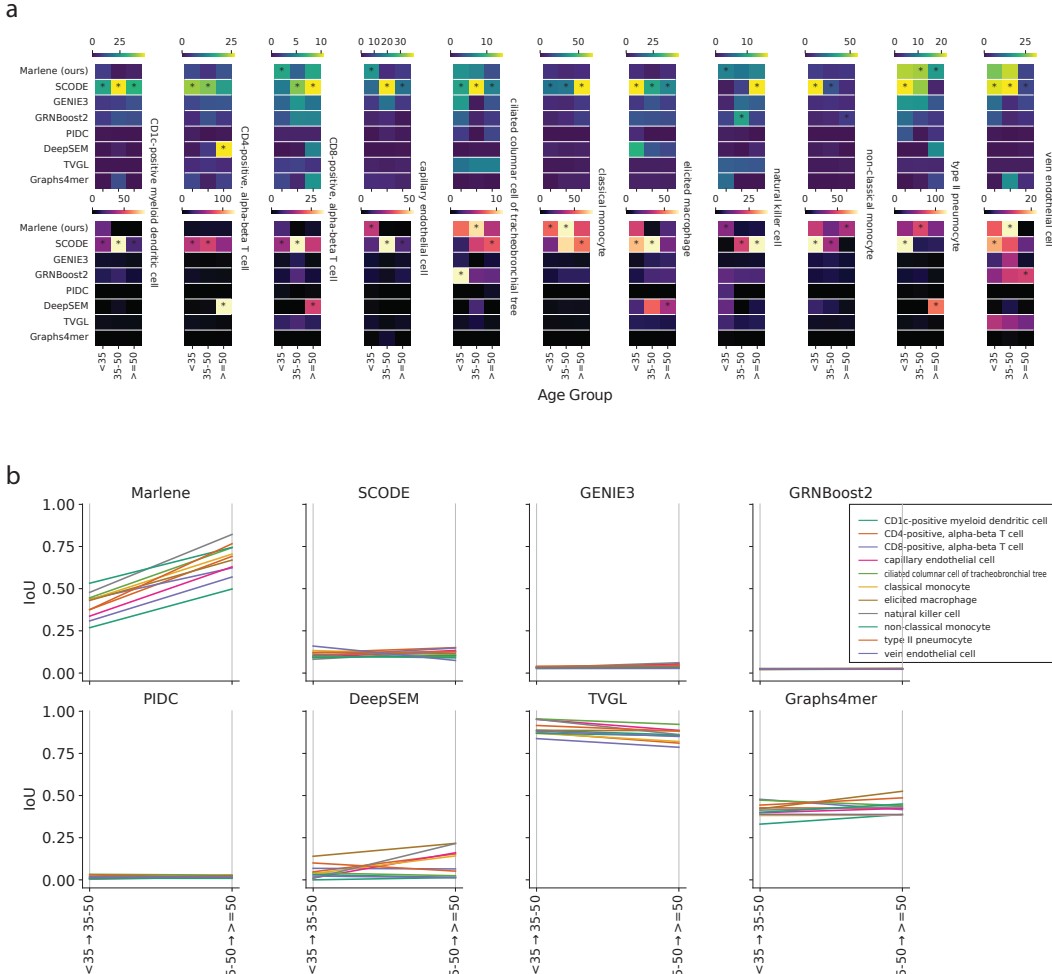

Figure 6: (a) FDR corrected $p$-values of Fisher exact tests reflecting the number of links that overlap with the two TF-gene databases. (b) IoU scores across time reflecting the overlap between consecutive graphs.

## A.4 ANALYSIS OF THE FIBROSIS DATASET

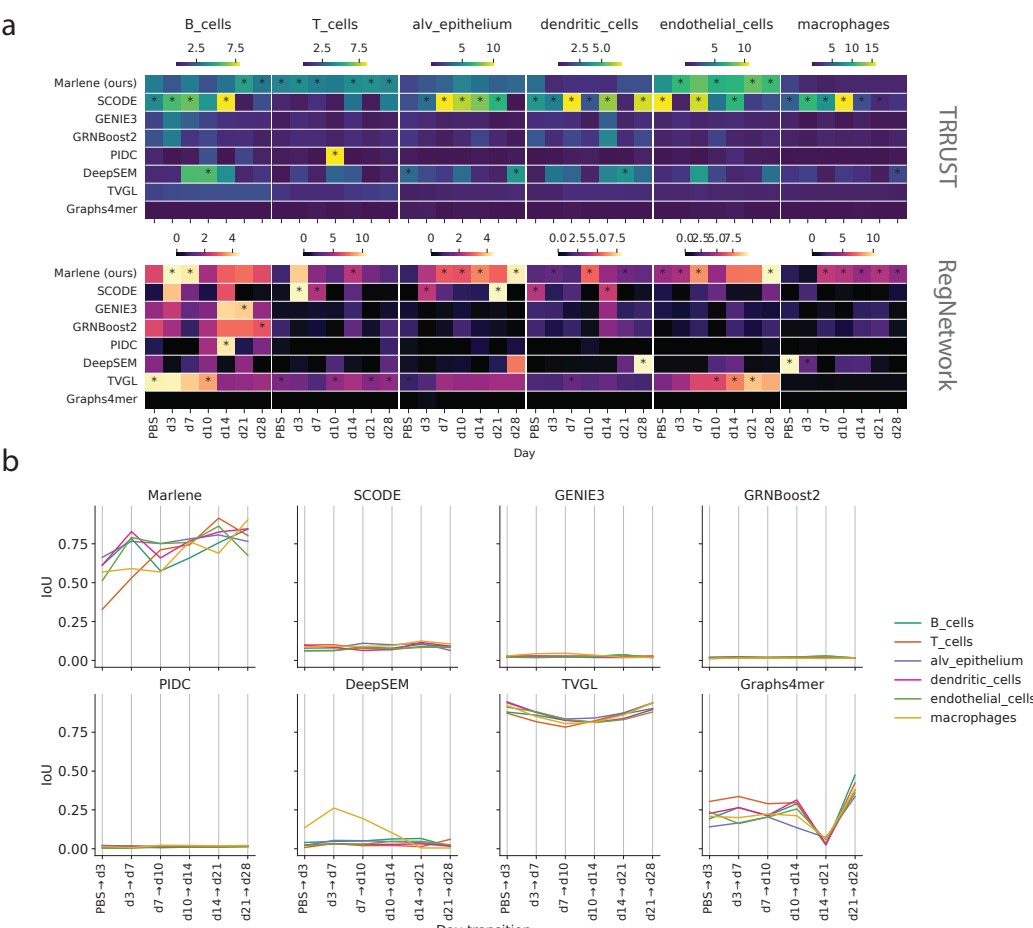

Figure 7: (a) FDR corrected $p$-values of Fisher exact tests reflecting the number of links that overlap with the two mouse databases. (b) IoU scores across time reflecting the overlap between consecutive graphs.

