# OpenReview forum: "Recovering Time-Varying Networks From Single-Cell Data"
_ICLR.cc/2025/Conference — ICLR 2025 Conference Withdrawn Submission_

### Official Review · Reviewer_m4Mi · 2024-11-01

**Soundness:** 3
**Presentation:** 2
**Contribution:** 3
**Rating:** 3
**Confidence:** 4

**Summary:**

The paper titled "Recovering Time-Varying Networks from Single-Cell Data" introduces Marlene, a deep neural network designed to infer dynamic gene regulatory networks from time series single-cell gene expression data. The authors propose a model that employs self-attention mechanisms and recurrent units to construct directed gene networks, with weights that evolve over time. A significant contribution is the integration of meta-learning to enable accurate recovery of temporal networks, even for rare cell types. The study demonstrates Marlene's effectiveness in identifying gene interactions relevant to specific biological responses across multiple datasets, showcasing the potential of deep learning in unraveling the dynamics of gene regulation and its implications for understanding complex biological processes.

**Strengths:**

1. propose a novel deep learning framework that employs self-attention mechanisms and GRUs to model the dynamics of gene regulatory networks.
2. The method is tested across three diverse datasets, which demonstrate the generalization ability of the model.

**Weaknesses:**

1. While the paper presents a technical approach, it could benefit from a deeper discussion on the biological implications of the findings and how they align with or differ from current scientific understanding.
2. In the experimental phase, mainly presents results based on metrics, demonstrating that it outperforms other methods. However, for solving a specific GRN problem, we are more concerned with whether the inferred GRN can identify some key genes or transcription factors (TFs) that can be further analyzed downstream. For a biological problem, merely comparing metrics does not adequately demonstrate the model’s performance.
3. The paper emphasizes using GRU to capture the temporal dynamics of the GRN. However, there are already many studies on dynamic GRNs, such as Dictys: dynamic gene regulatory network dissects developmental continuum with single-cell multiomics. I believe it would be more convincing to compare with these dynamic methods.
4. In GRN inference problems, perturbation experiments for some key genes are also an important downstream analysis. Including some perturbation experiments could help validate the accuracy of the inferred GRN.

**Questions:**

1. While the paper presents a technical approach, it could benefit from a deeper discussion on the biological implications of the findings and how they align with or differ from current scientific understanding.
2. In the experimental phase, mainly presents results based on metrics, demonstrating that it outperforms other methods. However, for solving a specific GRN problem, we are more concerned with whether the inferred GRN can identify some key genes or transcription factors (TFs) that can be further analyzed downstream. For a biological problem, merely comparing metrics does not adequately demonstrate the model’s performance.
3. The paper emphasizes using GRU to capture the temporal dynamics of the GRN. However, there are already many studies on dynamic GRNs, such as Dictys: dynamic gene regulatory network dissects developmental continuum with single-cell multiomics. I believe it would be more convincing to compare with these dynamic methods.
4. In GRN inference problems, perturbation experiments for some key genes are also an important downstream analysis. Including some perturbation experiments could help validate the accuracy of the inferred GRN.

---

> ### Author Response · Authors · 2024-11-13
>
> We thank the reviewer for the thoughtful comments. We address the weaknesses and questions below.
>
> 1. This is indeed an important discussion. However, in this manuscript, we evaluate our method by finding support in already established benchmarks and known biological processes. Investigating novel edges or processes in detail would require further mechanistic studies, which are beyond the scope of this paper. Our goal is to demonstrate the model's ability to recover known regulatory interactions effectively.
> 2. We address this concern through gene set enrichment analysis on human datasets, showing that genes added between time points are enriched for relevant biological processes that align with the underlying biological response. This suggests that Marlene understands the biological context of the sample.
> 3. We mention Dictys in lines 59-60. Basically, while this is a dynamic method, it cannot be applied to our setting as it relies on a multimodal approach (using both scATAC-seq and scRNA-seq types of data which is less prevalent and more costly). The GRN inference in Dictys is achieved by leveraging TF binding sites obtained from scATAC-seq, so it has more information than scRNA-seq.
> 4. Perturbation analysis would indeed be a valuable addition. However, as noted in lines 513-515 in the manuscript, our goal was not to perform causal inference or perturbation studies, but to uncover the underlying GRNs given a biological dataset. We see this as a promising area for future work.
>
> We hope these answer the reviewer's questions.

---

### Official Review · Reviewer_K927 · 2024-11-02

**Soundness:** 2
**Presentation:** 4
**Contribution:** 2
**Rating:** 3
**Confidence:** 4

**Summary:**

This paper proposes a novel deep learning framework called Marlene for reconstructing dynamic gene regulatory networks (GRNs) from time-series single-cell RNA sequencing (scRNA-seq) data. The authors aim to address the challenges of modeling the temporal evolution of GRNs using the increasingly available scRNA-seq data, which existing network inference methods are not well-equipped to handle.

Problem Significance:
Studying the dynamic regulation of biological processes is crucial for understanding the mechanisms driving responses such as development, disease progression, and treatment outcomes. Reconstructing accurate temporal models of GRNs can provide key insights to identify potential interventions and treatments. However, the increasing scale of scRNA-seq data and the presence of multiple cells profiled per time point pose challenges for traditional GRN inference approaches. The development of Marlene tackles an important open problem in the field.

Model Input/Output:
The input to Marlene is a time series of gene expression matrices {X^1, ..., X^T}, where each matrix X^t has dimensions (cells x genes) for a given time point t. The number of cells may vary per time point.
The output is a series of directed weighted graphs {G^1, ..., G^T} representing the GRN at each time point. The nodes are genes (assumed to be the same across time) and the edges capture regulatory relationships between transcription factors (TFs) and target genes. The graphs are characterized by adjacency matrices {A^1, ..., A^T}.
For evaluation, Marlene's inferred networks are benchmarked against curated TF-gene interaction databases (TRRUST and RegNetwork) using Fisher's exact test to assess overlap significance. The dynamics captured between time points are analyzed by computing intersection-over-union (IoU) scores.

**Strengths:**

Strengths:
* Marlene effectively leverages recent advances in deep learning, such as self-attention mechanisms and recurrent units, to model dynamic GRNs from scRNA-seq data. The approach uses set-based architectures to handle multiple cells per time point.
* Employing meta-learning (MAML) enables Marlene to reconstruct accurate networks even for rare cell types by treating cell types as tasks. This enhances the model's ability to handle heterogeneous cell populations.
* The model demonstrates strong empirical results on three diverse datasets, outperforming several static and temporal baselines in recovering known regulatory interactions and identifying relevant biological processes.

**Weaknesses:**

Weaknesses:
* The paper primarily evaluates using overlap analysis with existing incomplete databases of static interactions. More direct experimental validation of novel predicted regulatory links would strengthen the findings.
* Potential limitations in scaling Marlene to a very large number of genes are not thoroughly discussed. In experiments, the quadratic memory usage from adjacency matrices led to gene filtering.
* The model currently lacks the ability to predict the effects of perturbations like TF knockouts, which could enhance its utility for causal inference and treatment design.
* The usefulness of time-varying GRNs is not discussed.

Overall, the paper makes a valuable methodological contribution to a significant problem and demonstrates promising results. However, there is room for further validation and discussion of scalability limitations. More concrete insights into Marlene's novel interactions and their biological significance would also improve the impact. Overall, this is a solid paper that advances the field of GRN inference from single-cell data.

**Questions:**

1	How sensitive is Marlene to the choice of hyperparameters, such as the number of attention heads, hidden units, or depth of the neural network layers?
Explanation: The performance of deep learning models often depends on selecting hyperparameters. The paper does not provide a comprehensive analysis of how different hyperparameter settings affect Marlene's ability to recover accurate GRNs. Understanding the model's sensitivity to these choices is important for assessing its robustness and guiding practical applications.

	2	Can Marlene effectively handle datasets with a large number of time points or with irregular time intervals between samples?
Explanation: The study's datasets contain a relatively small number of time points (3-7). It is unclear how well the model would scale to longer time series, which are common in many biological processes. Additionally, the paper does not discuss how Marlene would handle irregularly sampled data, where the time intervals between consecutive points vary. Addressing these scenarios is crucial for the model's general applicability.

	3	How does the model's performance change when dealing with datasets of different sizes, both in terms of the number of cells and the number of genes?
Explanation: The paper reports results on three specific datasets but does not provide a systematic analysis of how the model's performance scales with the size of the input data. Understanding Marlene's data efficiency and ability to handle datasets of varying sizes is important for assessing its practical utility and guiding data collection efforts.

	4	How does Marlene handle technical noise and batch effects that are common in scRNA-seq data?
Explanation: Single-cell RNA sequencing data often contains technical noise and batch effects that can confound the analysis of biological variation. The paper does not explicitly discuss how Marlene deals with these issues or if any preprocessing steps (e.g., normalization, batch correction) were applied to the input data. Clarifying the model's robustness to these factors is vital for its reliable application to diverse datasets.

	5	Can the model provide insights into the strength and directionality of the inferred regulatory interactions?
Explanation: While Marlene outputs weighted directed graphs, the paper focuses primarily on evaluating the presence or absence of edges against existing interaction databases. It does not delve into how well the model captures the strength and directionality of the regulatory relationships. Providing a more detailed analysis of these aspects could enhance the interpretability and biological relevance of the inferred networks.

6    What are the downstream applications of time-varying GRNs? Currently, the authors evaluate according to the recovery of (static) curated interactions. How can the method be evaluated in a way that evaluates its utility for real downstream applications?

7   The paper shows an extremely large difference between Marlene and alternative methods. This is surprising because all methods perform largely the same tasks with only minor methodological tweaks. What can explain the large differences between models?

---

> ### Author Response · Authors · 2024-11-13
>
> We thank the reviewer for the thoughtful and detailed feedback. We appreciate your acknowledgement of the methodological contributions in our paper.  We hope to address the weaknesses below.
>
> 1. We appreciate the suggestion for direct experimental validation. In this manuscript, we evaluate our method by comparing it to well-established benchmarks and known biological processes. Investigating novel edges or processes in detail would require further mechanistic studies, which are beyond the scope of this paper. Our goal is to demonstrate the model's ability to recover known regulatory interactions effectively.
> 2. Scalability was not the primary focus of this manuscript. We discuss in the Limitations section that more efficient implementation (e.g., using FlashAttention) would lead to a better capability of the model to handle larger numbers of genes. We mainly view the current manuscript as a methodological contribution, but we would be happy to extend the method further to accommodate larger datasets.
> 3. Perturbation analysis is important, however, as noted in lines 513-515 in the manuscript, this is a different task. We were mainly concerned with uncovering the underlying GRNs from time series data which is already a significant goal. Causal inference and knockouts were not out goal and it would require other types of data, such as interventional data.
> 4. We thank the reviewer for highlighting the need to discuss the usefulness of time-varying GRNs. Since most biological processes are dynamic, time-varying GRNs are necessary to understand the evolution of regulatory interactions in the cell. For instance, understanding what TF-gene links are disrupted could help researchers discover drugs targets for specific TF-gene connections. We will expand our motivation section to add applications of time-varying GRNs.
>
> Answers to the reviewer's questions can be found below.
>
> 1. We minimally tuned the parameters in this study (mainly learning rate), favoring significance of the results. While we do not use a hyperparameter tuning approach in this work, we believe that the results will improve with proper tuning.
> 2. Due to cost, most scRNA-seq datasets contain few time points, so we do not necessarily view this as a limitation of the method. However, other temporal units such as Structured State Space models (S4), can help with longer time series. Regarding the irregularity of time points, some of the datasets we used in the experiments contained irregular time points. For instance, the COVID-19 dataset contained time points from days 0, 2, 10, 28. Meanwhile, the time points in the aging dataset were separated by several years of age.
> 3. The number of cells of a single type varies greatly for each of the datasets (some only contain a few hundred, other thousands). Some types contained few cells, which is where meta-learning helps. We do not report the breakdown of the datasets by cell type, but we will add this information to the supplement.
> 4. Predicting cell type rather than exact gene expression is one way to mitigate this problem. The lung aging dataset is constructed by combining several datasets that come from different sources, thus it is quite prone to batch effects. However, as we show, Marlene performs better than most baselines for this dataset.
> 5. Connection strength could be correlated with the attention weights, however, it is unclear how meaningful the precise weights are for these matrices (interpretability of attention weights is limited). Therefore, we prefer to take the top x% of the edges and study them as a set, rather than looking at the weight. Directionality of the regulatory link is given by the attention (these matrices are not symmetric) and we analyze them as directed tuples in the results section.
> 6. In addition to static interactions, we also include gene set enrichment analysis of dynamic genes (i.e., genes that change between time points). This provides insight into the dynamic biological response and could, for example, help drug discovery efforts to target the right TFs.
> 7. Many of the baseline methods are static. Therefore, the lack of dynamic modeling might help explain some of the discrepancy. Also, the modeling scheme is different, as most of these baselines are not based on highly expressive nonlinear networks, but rely on simpler shallower approaches.
>
> The reviewer has praised the work, and their main concerns were with additional experimentation or extensions of the work. Given this and the comments, we feel like the score of 3 is too low for this work. We would appreciate it if the reviewer could increase their score.

---

### Official Review · Reviewer_mAx7 · 2024-11-04

**Soundness:** 4
**Presentation:** 3
**Contribution:** 4
**Rating:** 8
**Confidence:** 4

**Summary:**

This paper introduces an attention and graph based approach called Marlene, to infer dynamic GRN from time series (longitudinal) single cell RNA sequencing data. The author provided an in-depth overview of the GRN inference field and explained the need of developing novel methods to capture the time-varying gene networks.

Marlene starts with a pooling by multihead attention (PMA) layer which transform gene expression profiles at multiple time points to gene feature matrix at multiple time points. Then self attention are applied on each of these gene embeddings and the adjacency matrix is calculated based on the learned transformation. The weights of the self attention module are connected with GRU units so that information could be shared across time points. Finally, the extracted GRN are used to regenerate the expressed data, and the generated features are used to predict the cell type label. The entire model is trained to predict cell type.

In term of experiment, the author managed to squeeze results from 3 experiments into this paper. These 3 cases are SARS-CoV-2 Vaccination, Aging and Lung, and fibrosis a mouse lung injure model. All 3 experiments are solid and supporting the claim.

**Strengths:**

This is a very solid piece of research. The motivation is well explained and attractive. The idea is novel and has potential to be practically useful. The description of the method is very clear and the logic flows very well. The experiment part is comprehensive.

In terms of the method itself, using PMA or Set transformer to convert the expression to gene feature matrix eliminate the axis of cells so downstream analysis could focus on the attentions on genes. The use of GRU in the next step is not so intuitive but seems to have literature support. The task of predicting cell labels is also very clever in this case because single cell data is noisy and complete reconstruction is more prone to error. Also, control the sparsities of the adjacency matrix by using the top k edges is also very inspiring.

**Weaknesses:**

1. I would like to see a more clear explanation on Equation 3-6. What exactly are the rational of using GRU here beyond it was used in EvolveGCN? Do we have any physical meaning on this operation on Equation 3-6?
2. Algorithm stability is a key metric in BEELINE. Could you comment on the stability of Marlene?
3. The output of Equation 6 is the adjacency matrix at timepoint t. Then, at least for the evaluation you have performed in this study, you must have transformed multiple At into one At. Could you explain in detail how you did that?

**Questions:**

1. BEELINE does have a few dataset that has multiple time points (For example, hESC). I wonder how Marlene perform on those datasets in BEELINE. You can use the non-chipseq ground truth, which is very similar with the method I use. It would be great if you can use their metric (EPR/AUPRR etc)

---

> ### Author Response · Authors · 2024-11-13
>
> We thank the reviewer for praising the work. We hope to address their concerns below.
>
> 1. We apologize for the lack of clarity on this point. The connection to EvolveGCN is the dynamic adaptation of weights. In most neural networks, weights are updated via some gradient-based algorithm. In EvolveGCN and Marlene, some of the network weights are predicted by the model itself. In the case of Marlene, the GRU module predicts the weights of the linear layers in the self-attention module for the next time point. These weights are then multiplied with the input x to output keys, queries. We hope this clarifies the use of the GRU.
> 2. This is a good point. As is common in deep learning literature, neural networks are sensitive to initialization and this is also true for Marlene. However, we found that many TFs and genes were consistently selected across runs.
> 3. Good catch; this is not explained in the paper. During evaluation, we obtain $A_t$ for multiple batches and then average the outputs to obtain a single $A_t$ for each $t$. We have now clarified this in the main text.
>
> We thank the reviewer for suggesting the additional BEELINE datasets. We looked at BEELINE datasets from GEO # GSE75748. However, these datasets are derived from a single cell type, hESCs. Currently, Marlene is designed to predict the cell type which we claim helps with the discovery of cell type specific GRNs. In the case of hESCs, there is only one cell type, so this data does not leverage the benefits of meta learning or Marlene for cell specific GRNs.
>
> We hope this clarifies some of the reviewer's comments.

---

### Official Review · Reviewer_4vp2 · 2024-11-04

**Soundness:** 2
**Presentation:** 3
**Contribution:** 2
**Rating:** 3
**Confidence:** 4

**Summary:**

The paper aims to predict a series of graphs describing gene expression regulation by transcription factors from a series of single-cell gene expression data. The method uses attention-based architecture within each time point, with a recurrent component linking the time points.

**Strengths:**

The paper focuses on an important, though well-studied task of regulatory gene network inference. It focuses on time-series, single-cell data, an increasing available, more detailed view of gene expression.

The approach goes beyond simple application of existing deep learning models by using an architecture in which the projection matrices for calculating attention are evolve as part of an RNN.

**Weaknesses:**

The architecture relies on interpreting the attention matrix A generated within the model as the adjacency matrix of the regulatory network, with the model itself being trained on a surrogate task of predicting cell type (y). The assumption that A, used in this surrogate task, will capture direct regulatory interactions is not very well justified in the manuscript. Would using a matrix with different dimensionality (e.g. having #columns the same, to match # of TFs, but with different number of rows), result in similar performance on the surrogate task? If yes, what beyond shape leads to the interpretation of A as the adjacency matrix? For example, is there a reason to apply softmax to rows of the matrix?

The experimental results are missing key details relating to the performance on regulatory network inference. Key statistics related to network-wide performance in discovering edges are not reported: it would be helpful to see AUROC and AUPR values.

**Questions:**

The model is aimed at time series, but the experimental data very few data points. If the method was applied to each time point separately (as just the first time point, eliminating the recurrent part), would the performance be affected substantially?

---

> ### Author Response · Authors · 2024-11-12
>
> We thank the reviewer for their comments. The points raised regarding the interpretation of the adjacency matrix are valid, and we address them below.
>
> First, we acknowledge that attention weights do not inherently capture regulatory interactions. The limitations of interpreting attention weights have also been detailed in papers like "Attention is not Explanation" [1]. However, this is why we believe the softmax + reconstruction step add a level of biological plausibility. The softmax plays a double role: 1) it enforces a probabilistic interpretation. Each gene's regulatory influence is weighted across all TFs, i.e., the TF with most influence would be assigned a higher weight. 2) Most importantly, the softmax does not allow negative weights. Indeed, if the weights are close to 0, then they contribute minimally in the reconstruction step, otherwise, the larger the magnitude, the more effect this particular link will have in the reconstruction step. Having negative weights on top of the complex nonlinearities in the model, would make the interpretation of such weights very complicated. In short, we believe that the attention can be interpreted as an adjacency matrix because it is directly being treated as one through a message passing step during reconstruction.
>
> We agree with the reviewer that interpreting the exact values of these weights is limited, which is why we only take the top x% of the edges as the most influential and discard the specific weight. Given this, the computation of AUROC and AUPR values is also not very informative or meaningful. For this reason, we prefer direct overlap tests, like Fisher's exact test. If, instead, the suggestion is to compute these metrics by increasing the number of selected edges, this is also problematic as the set of ground truth edges is very small. Hence, increasing the number of selected edges would simply increase false positives.
>
> Reducing the number of rows in A (i.e., deleting genes) would impact the performance of gene enrichment analysis. We showed in the results section that the genes selected by the method are relevant to the response being studied (and most of these gene sets are small), suggesting that the relevant genes are indeed picked by the method.
>
> The surrogate task of predicting cell type was chosen in order to align the regulatory interactions with specific cell types which is the goal of the method.
>
> Finally, regarding the question on few time points, these are typical for scRNA-seq data. Most datasets do not extend beyond a few time points, which is one of the ongoing challenges in the field. Regarding the suggestion to run the method independently for each time point, we note that, for the very first time point in each series, there is no influence from the future. Despite this, the overlap with the ground truth databases is still very significant as shown in the figures. Hence, treating Marlene as a static method should still recover relevant edges. It is unclear, however, if the model will still capture the dynamics of the system (i.e., the consecutive edge sets may not be related to each other).
>
> We hope these address the reviewer concerns. Please let us know if you have additional questions.
>
> [1] https://arxiv.org/abs/1902.10186

---

### Note · Authors · 2024-11-18

I have read and agree with the venue's withdrawal policy on behalf of myself and my co-authors.